# Invisible ECG for High Throughput Screening in eSports

**DOI:** 10.3390/s21227601

**Published:** 2021-11-16

**Authors:** Aline Santos Silva, Miguel Velhote Correia, Hugo Plácido Silva

**Affiliations:** 1FEUP—Faculdade de Engenharia da Universidade do Porto, 4200-365 Porto, Portugal; 2INESC TEC/FEUP—Faculdade de Engenharia da Universidade do Porto, 4200-365 Porto, Portugal; mcorreia@fe.up.pt; 3IT—Instituto de Telecomunicações, 1049-001 Lisboa, Portugal; hsilva@lx.it.pt

**Keywords:** invisibles, off-the-person, electrocardiography, pervasive sensing, eSports

## Abstract

eSports is a rapidly growing industry with increasing investment and large-scale international tournaments offering significant prizes. This has led to an increased focus on individual and team performance with factors such as communication, concentration, and team intelligence identified as important to success. Over a similar period of time, personal physiological monitoring technologies have become commonplace with clinical grade assessment available across a range of parameters that have evidenced utility. The use of physiological data to assess concentration is an area of growing interest in eSports. However, body-worn devices, typically used for physiological data collection, may constitute a distraction and/or discomfort for the subjects. To this end, in this work we devise a novel “invisible” sensing approach, exploring new materials, and proposing a proof-of-concept data collection system in the form of a keyboard armrest and mouse. These enable measurements as an extension of the interaction with the computer. In order to evaluate the proposed approach, measurements were performed using our system and a gold standard device, involving 7 healthy subjects. A particularly advantageous characteristic of our setup is the use of conductive nappa leather, as it preserves the standard look and feel of the keyboard and mouse. According to the results obtained, this approach shows 3–15% signal loss, with a mean difference in heart rate between the reference and experimental device of −1.778 ± 4.654 beats per minute (BPM); in terms of ECG waveform morphology, the best cases show a Pearson correlation coefficient above 0.99.

## 1. Introduction

eSports is the organised competition of players or teams within video gaming. It is increasingly focused on performance and sporting success, with many professional players receiving a similar level of training to that of elite traditional sports athletes [1]. eSports has steadily entered mainstream entertainment in the last decade, and revolves around players or teams working to beat their counterparts often in a series of objectives. Popular eSports games such as League of Legends or Counter Strike: Global Offensive enjoy viewership in the many tens of millions at their premier live tournaments, with figures that rival traditional sports [2]. Revenue streams into eSports have also ballooned, surpassing $1 billion in 2019 [3]. This has trickled down to major increases in spending within professional teams on players, coaching, and facilities to improve performance. The staff that support the players have also expanded to include multiple managers, coaches, physiotherapists, sports psychologists, and data analysts.

A number of studies have already scratched the surface of performance determinants within eSports. Unsurprisingly, for team games, communication and collective intelligence are shown to be predictors of success [4]. Teamwork is critical at a professional and casual level within eSports [5], expanding to the point where players may not be selected or are even fired from a team, if there are weaknesses in these domains. Indeed, these qualities may translate into the game through measurable and significant outcomes, predictive of success, such as fewer deaths and better map control [6]. For example, fewer deaths mean that the team generally has a higher chance of winning decisive battles through higher damage or player numbers, whereas map control often forces the opposing team into disadvantageous positions. With the rise of eSports, specialized monitoring technologies and clinical grade assessments have become a topic of growing interest.

Smart watches and fitness trackers, for example have a large market value that is forecast to grow to $62 billion by 2023 [7]. As adoption increases, demand for functionality has broadened from heart rate monitoring to heart rate variability and sleep analysis. It was identified as early as 1998 that non-invasive devices can not only consistently display meaningful physiological data, but also serve in situations where traditional measuring tools are not viable [8]. The technologies are also employed in a number of professions and sports, with heart rate monitoring shown to be an effective marker of stress in pilots (e.g.) during training [9]. The military has also embraced physiological sensing technology as a method of improving performance by collecting information on soldier fitness, alertness and their psychological state [10]. Formula One racing could be one of the earliest adopters of physiological sensing in sports, due to the increasing importance of the physical and mental strain applied on the drivers [11,12].

As the rewards for success increase alongside knowledge of the stressors within eSports, monitoring devices are expected to play a key role in identifying and improving outcomes. Nevertheless, existing body-mounted devices may interfere with the gameplay and be perceived by the players as a distraction. This article discusses the potential value of physiological monitoring within eSports, the sensing technologies currently found in the state-of-the-art, the barriers that must be overcome to ensure full implementation, and proposes an “invisible” sensor integration approach applicable to eSports performance assessment.

With the goal of extending the state of the art for devices that do ECG acquisition, our work addresses issues related to electrode materials, sensor integration into the keyboard and mouse, and by adopting a sensor design that requires fewer contact points between the sensor and the body. The rest of the paper is organized as follows. Section 2 describes the background and the state of the art. Section 3 details the implementation. Section 4 summarizes the experimental evaluation and results. Finally, Section 5 outlines the main conclusions and future work direction.

## 2. Related Work

Physiological data has been found to be of added value in multiple dimensions of competitive eSports games, due to the associated stress, complex decision making, teamwork, and even duration of activity [13,14,15,16]. Nowadays, there is a plethora of personal monitoring devices, which can measure variables such as heart rate (HR), brain activity, skin conductance, or temperature, just to name a few. One of the most commonly utilised portable technologies are smartwatches and fitness trackers. These devices have grown in variety with increasing functionality over successive generations. It is now common for physiological parameters such as HR, temperature, and physical activity levels to be measured, often with the aim of improving health outcomes ranging from self-management of diabetes to seizure monitoring fast pace of development [17]. Similarly, sleep trackers are increasingly common place and are a cost effective tool for monitoring sleep, though they appear to vary in reliability [18,19]. In the context of eSports, there is a natural applicability of these devices in training, especially considering the setting of a gaming house; an arrangement in which players cohabit a tailored facility with access to high speed internet and quality computing equipment. In this environment, many factors are often more controlled such as sleep and activity cycles.

Despite high levels of investment, few of these devices have managed to reach clinical grade accuracy or reliability [20]. Clinically focused wearables, on the other hand, have generally completed longer periods of device evolution with more stringent standards for approval and application. These technologies are often based on known physiological principles adapted for utility under controlled conditions. Whilst this makes the readings more reliable, the devices themselves are more difficult to exploit outside of their limited tested environment. Furthermore, a lack of awareness and the generally higher cost make these options less accessible. As set forth by the state-of-the-art, both general consumer and clinically focused technologies offer value to eSports [13,14,15,16].

The devices range from a monitor placed on the wrist, to sensors distributed within clothing. Importantly, these devices should in theory allow the player to compete as normal without hindrance. The diversity of tools enables players and teams to work in a minimally restrictive manner whilst maximising analytical potential, with scope to combine technologies, expanding monitoring capability further. Notably, portable devices frequently utilise technologies that can derive multiple physiological parameters such as bioimpedance (a measure of electrical resistance through the body), to obtain heart rate and respiratory rate with reasonable levels of accuracy [21,22]. This is useful, as playing video games can increase psychophysiological arousal manifesting in parameters such as heart rate variations, suggesting that the cognitive and subsequent physical load can be significant [23]. The association of cardiovascular activity with exercise and mental demand is well documented; heart rate is one of the most commonly assessed physiological parameters, and varies with both physical and cognitive exertion [24]. It is often treated as a surrogate for sympathetic activity and can be helpful to assess stress.

In fact, HR is the base for an extensive set of analysis that are generally described as Heart Rate Variability (HRV). This is a group of psychophysiological data analysis techniques that builds upon the inter-beat changes that occur either naturally in the heart, or as a response to a stimuli/mental demand state. It has been frequently associated with exercise and even mortality in clinical population [25,26]. HRV appears to vary within video gaming and may be related to a variety of factors such as the activities within the game, the content shown, or difficulty of the task [27]. Furthermore, work by Lee et al. [28] suggests that HRV will also differ based on the background of the player. Participants fulfilling the criteria for internet gaming disorder more often have suppression of high frequency HRV, a phenomenon that is associated with parasympathetic inhibition and emotions such as stress or anxiety [29]. The stroke volume (SV) is the volume of blood pumped from the heart per beat. The SV can be multiplied by the HR to derive the cardiac output (CO) which is the volume of blood pumped by the heart in one minute, and is linked to exercise capacity. Notably, the SV and CO have been shown to increase with higher cognitive load and exercise [30,31]. Previous research has also found associations between changes in the heartbeat waveform morphology and the attention level of subjects during computer use, which may also be applicable to eSports [32]. While the 12-lead electrocardiogram (ECG or EKG) can be employed, this is rarely practical outside of the hospital setting, especially in the context of body movement. Portable devices with ECG capability are common, and generally utilise 1–3 leads making it viable for general monitoring. Despite this potential, there are well-known barriers to the application of existing wearable and body-worn technologies in the field of eSports [33], chief among which are the interference with gameplay and discomfort. These motivate the need to develop alternative sensing solutions, ideally more “invisible” to the players [34].

## 3. Proposed Approach

### 3.1. Methodology

Our proposed approach consists of a full-scale model of a gaming keyboard armrest and mouse, Figure 1, with the main areas that come in contact with the skin wrapped in a conductive nappa. The sensor and data acquisition system is the same used in [35], allowing the keyboard armrest to transmit the collected signals via Bluetooth to a receiver (i.e., a computer or a smartphone).

In parallel with the keyboard armrest, data is simultaneously collected with a second system applied to a reference location on the bodies of the tested subjects (here considered to be the gold standard). The experimental setup also included a microphone sensor on the gold standard and a Buzzer on the keyboard stand, to allow synchronization of both independent time series in post-processing. This acoustic approach was adopted to ensure electrical decoupling between systems.

We used Python 3.8, the BioSPPy library (0.6.1) [24] for the digital filtering and segmentation methods, in order to analyze and characterize the signal, and in order to analyze the heart rate variability (HRV) PyHRV (0.4.0) was used. The mean values and standard deviation were calculated over all subjects.

### 3.2. Data Acquisition

For benchmarking purposes, we collected data simultaneously using the gold standard system (ECG REF) and our proposed approach (ECG EXP). Disposable 24 mm diameter H124SG KENDALL ARBO surface electrodes were used with the ECG REF system, and the electrode leads were applied to the subject in an Einthoven’s Lead I equivalent configuration; as shown in Figure 2, the lead REF was applied on the cervical—C5/C6, the IN- terminal on the right clavicle, and the IN+ terminal on the left clavicle. With the ECG EXP system, we used dry electrodes composed of conductive napa leather, applied to the keyboard armrest and computer mouse. This setup produces a total of four time series, namely the buzzer triggering (O1), the ECG REF time series, the acoustic sensor response to the buzzer (MIC), and the ECG EXP time series.

Informed consent was obtained from all participants for this study following a protocol whose experimental methods are in accordance with the guidelines and ethical principles of research involving human subjects established by the Declaration of Helsinki, and were submitted to and approved by the IT—Telecommunications Institute ethics committee. There were a total of 7 healthy volunteers aged 24–34 years enrolled; of these participants, 6 were female and 1 had androgenic body hair. For each participant, 15 min of data was recorded while sitting with skin in contact with the electrodes.

We refer the interested reader to [35], for more details on the signal post-processing steps. However, using methods included in the BioSPPy Python module, the signals are filtered using a 45 Hz Finite Impulse Response (FIR) filter of order 30, and segmented using the method proposed by Hamilton [34]. Before comparing the ECG EXP with the ECG REF, we ensure that only matched segments are used, i.e., due to the influence of noise, some QRS complexes of ECG REF may not have valid matched QRS complexes in ECG EXP (and vice versa). Heart rate analysis is only performed for segments in which two or more R-peaks are available (allowing heart rate calculation in both signals). Comparison of the complete waveform morphology (i.e., P-QRS-T waves) is also performed only for the matching segments in both ECG REF and ECG EXP signals.

## 4. Results

### 4.1. Skin-to-Electrode Impedance

Some factors such as age, temperature, humidity, moisturizers and sun exposure can cause an influence on the impedance of the skin, which generates a change in the quality of the ECG signal. This impedance can be described as the resistance of alternating current electrical signals to pass through, and is measured as the voltage/current ratio. The skin becomes a part of the circuit when the electrode is in contact with the skin surface. If there is a high skin impedance, it causes the signal to be compromised, originating a drop in signal quality which generates substantial noise and even signal loss, due to the isolation of the skin-electrode interface. In order to determine the impedance, a circuit was created using a 1k resistor R, two Ag/AgCl wet electrodes(+OV), and the experimental electrode for which we want to determine the impedance in contact with the skin (Figure 3, Table 1). With the oscilloscope we determined the maximum amplitude value for each channel, *VA* and *VB*. From these values the impedance magnitude was determined as per Equation (Equation 1).
(1)$$Z\left(k\Omega \right)=\frac{R\times VA}{VB}$$

### 4.2. Rhythm Analysis

With this analysis we aim to characterize the potential differences in heart rate as calculated from signals collected with ECG REF and with ECG EXP. Distortions in the R-peak can introduce latency that affects heart rate calculation, and artifacts in the signal can lead to undetected or erroneously detected peaks.

A summary of a comparative statistical analysis is shown for the heart rate having its mean (μ) heart rate along with the standard deviation (σ). In addition a comparison of the heart rate difference between each ECG EXP channel and the ECG REF channel for each cycle pair of R-peaks is presented. The percentage of noisy segments is also presented, corresponding to periods when the signal is saturated, or highly corrupted by noise. The signal detection error (SDE in %) is given by Equation (Equation 2), where *S* represents the total signal and *N* represents the signal outside the measurement range (both in seconds), Table 2. For this case we will have ECG EXP when both arms are in contact with the keyboard stand and the other case is when one arm is in contact with the keyboard stand and the other hand is in contact with the mouse.
(2)$$\mathrm{SDE}\left(\%\right)=\frac{\left(S-N\right)}{S}$$

The HRV analysis was conducted by comparing the R-R intervals, the Poincaré analysis (a geometric technique used to evaluate long-term data scattering in RR time series), and the Detrended Fluctuation Analysis (DFA). The Poincaré analysis is based on the derivation of the standard deviation parameters along the minor axis, major axis, by the ratio of both and the area of the eclipse, respectively SD1, SD2, SD1/SD2 (Equation (Equation 7)) and s(Equation (Equation 6)). The SD1 parameter (Equation (Equation 3)) is calculated using the time domain parameter of the standard deviation of successive differences (SDSD). The SD2 parameter (Equation (Equation 4)) is calculated using the standard deviation of successive differences (SDSD) and standard deviation of NN series parameters (SDNN) parameters.
(3)$$SD1=\sqrt{\left(1/2.SDSD^{2}\right)}$$
(4)$$SD2=\sqrt{\left(2.SDNN^{2}-1/2.SDSD^{2}\right)}$$
(5)$$SD_{ratio}=SD1/SD2$$
(6)s=π.SD1.SD2

The DFA non-linear dynamics analysis is typically used in HRV to analyze correlations of NN time series, with its origins being based on the definition of self-affine processes. For a given process X, if the standard deviation (s) of the values within a window of length n, changes with the window length factor L in a power law (Equation (Equation 7)), the process is said to be self-affine.
(7)$$\sigma \left(X,L\times n\right)=L^{H\times \sigma (X,n)}$$
the standard deviation of the process X, computed using windows of size k, is described as σ(X,k). The paramente H found in the equation is typically known as the Hurst parameter. Like the Hurst exponent, *H* is obtained from a time series calculated by F(n), or σ(X,n), for different *n*, and fitting a straight line to the plot of log(F(X,n)) versus log(n). When computing a single F(X,n), the time sequence is partitioned into windows of equal size *n*, so that the ith window of this size has the form Equation (Equation 8).
(8)$$W\left(n,i\right)=\left[x_{i},x_{i+1},x_{i+2},...,x_{i+n-1}\right]$$

Calculating s(W(n,i)) for each i, and averaging the resulting values over i, enables us to obtain s(X, n). Table 3 and Figure 4 show an example of the Poincaré and DFA data for a subject (randomly selected). While the ECG REF and ECG EXP utilizing the keyboard contact show comparable trends, the ECG EXP compared to the keyboard/mouse shows a greater amount of previous NN intervals. These findings further reinforce that the performance is comparative to the gold standard, and is thus the next data source to consider for further analysis.

### 4.3. Heartbeat Waveform Morphology

The morphology of heartbeat waveforms is an important component of ECG analysis. As such, with this analysis we seek to assess the point-by-point morphological similarity between the heartbeat waveforms obtained using ECG REF and ECG EXP. For the segmentation of the heartbeat waveforms, the R-peaks of the ECG REF are detected first; afterwards, a decision criterion was used for the detection and removal of outlier heartbeat waveforms. These steps are done as per the method described in [35].

In order to illustrate the individualized heartbeat waveforms, in Figure 5 we visualize the heartbeat waveforms considered valid represented in yellow and the heartbeat waveforms considered outlier represented in gray. Furthermore, in order to represent the statistical analysis of the waveforms obtained with the materials that showed better performance, the Table 4 presents the values of Pearson’s correlation coefficient and Normalized Root Mean Square Error (NRMSE). Based on the results of Section 4.2, which were subsequently confirmed experimentally, it can be proven that the signals obtained by the proposed device (ECG EXP) are correlated with the reference signals (ECG REF).

### 4.4. Effect of Skin Moisturizer

When the electrode is placed on the skin surface, the skin becomes an integral part of the circuit. If this circuit is compromised due to high skin impedance due to factors such as age, sun exposure, skin lotions, relative humidity, and ambient temperature, signal quality can be negatively affected, causing loss of baseline, substantial noise (e.g., motion artifacts), and even loss of signal (e.g., due to isolation of the skin-electrode interface).

In this respect, the use of high or low density skin moisturisers is another influencing factor. To assess this aspect, a test was carried out, in which one subject used two types of moisturisers, one with low density and the other with high density, and the signal was acquired using the keyboard electrode, which presented the best results. As shown by Figure 6 and Table 5, when in contact with moisturizer A, of low density, it was still possible to obtain an ECG signal, even though it showed some disturbances. When moisturizer B was used, we can observe that the signal morphology is significantly distorted. These results suggest that changes in the skin impedance due to skin moisturizer ultimately affect the ECG signal.

## 5. Conclusions

eSports have grown with the increase in available technologies, making the concept of live monitoring of players during training and games not only feasible but also potentially mutable. Our research further contributes to the state of the art in invisible ECG, i.e., a way of collecting physiological signals embedded in users’ daily lives, by exploring industrially viable electrode materials that can be produced and effectively integrated into a keyboard holder and suggests that there is a growing overlap between clinically focused devices and general consumer products with clinical-grade measurement capabilities, making this an exciting area for future studies.

To acquire the ECG signals on the keyboard armrest, we used dry electrodes with a conductive nappa. According to the results obtained, they showed acceptable results, as demonstrated by the experimental evaluation performed using heart rate, HRV, and the morphological analysis of the collected ECG signals. As demonstrated in the Section 4.2 and Section 4.3, the experimental results confirmed that the conductive nappa provides an adequate electrical interface with the skin in most subjects.

A prototype of an instrumented keyboard armrest and mouse was created, which aggregates the technical solutions that demonstrated the best performance and form during the design and development process. Future work will focus on exploring the evaluation of users with known pathological conditions and different age groups. However, this work further strengthens the feasibility of ECG data acquisition at the arms/fingers using electrodes next to a conductive nappa embedded in a surface that eSport players interact with on a regular basis.

## Figures and Tables

**Figure 1 sensors-21-07601-f001:**
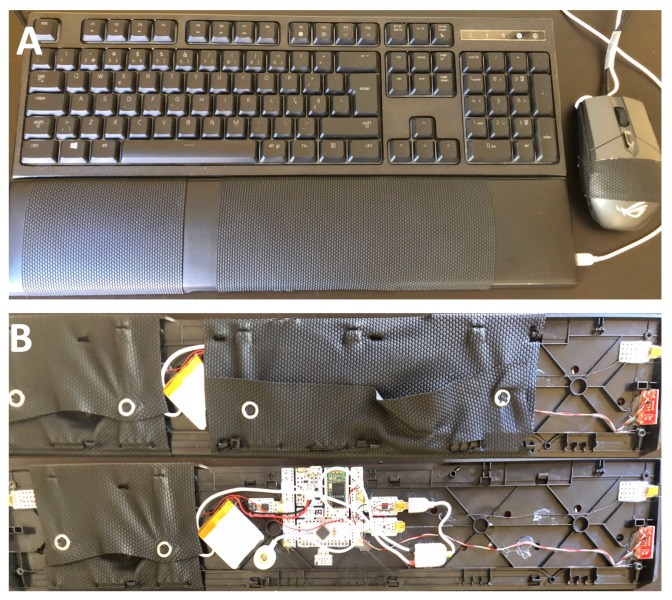
Prototype of the keyboard armrest, highlighting (**A**) the positioning of the electrodes and (**B**) the data acquisition system.

**Figure 2 sensors-21-07601-f002:**
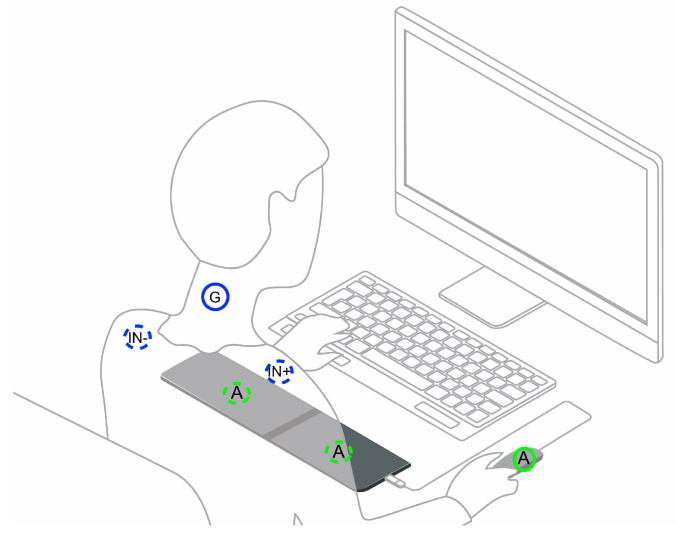
Experimental setup showing the electrode placement for the gold standard system (ECG REF: IN+, IN- & Ground) and the experimental electrodes location (ECG EXP: A).

**Figure 3 sensors-21-07601-f003:**
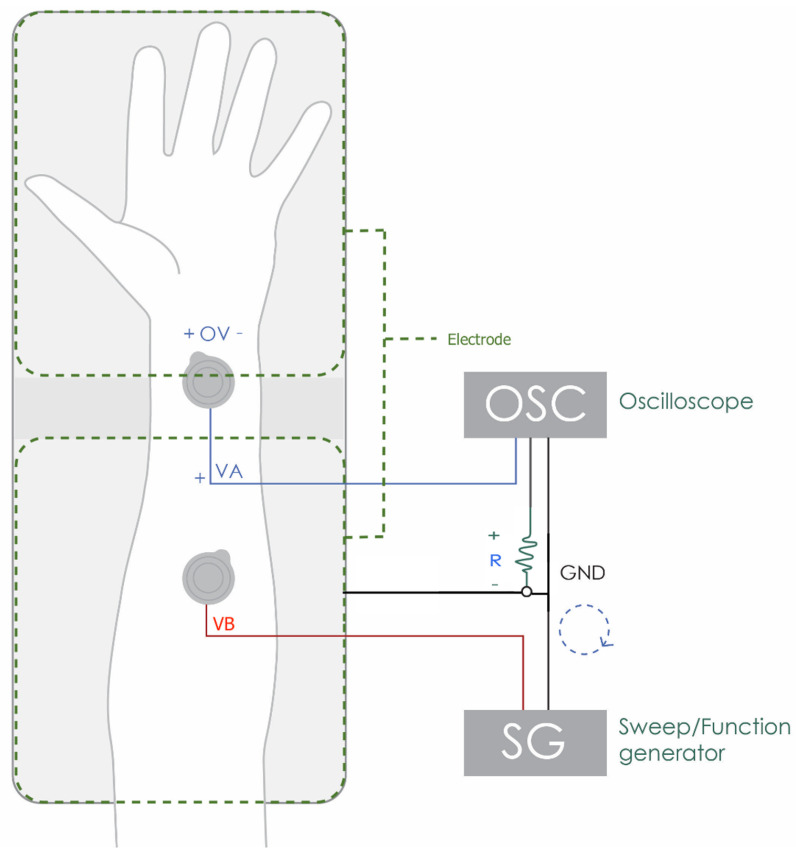
Experimental setup for skin-to-electrode impedance measurement.

**Figure 4 sensors-21-07601-f004:**
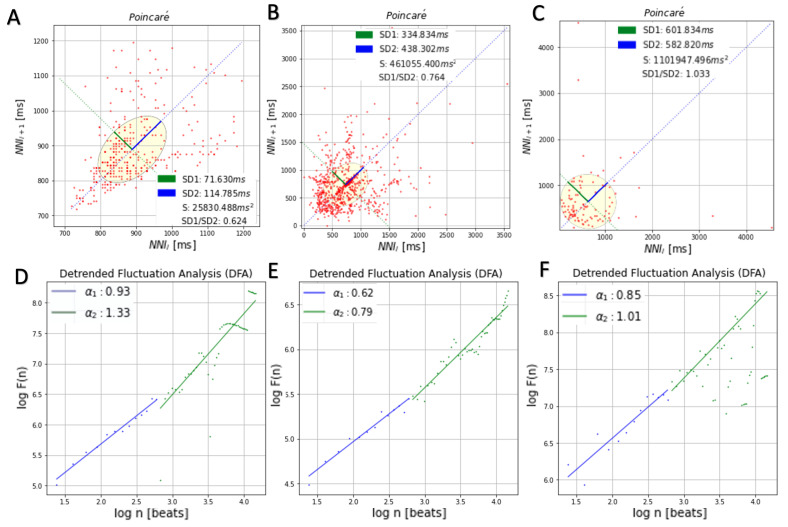
Poincaré and DFA plots for ECG REF (**A**,**D**) and ECG EXP with keyboard (**B**,**E**) and keyboard/mouse (**C**,**F**).

**Figure 5 sensors-21-07601-f005:**
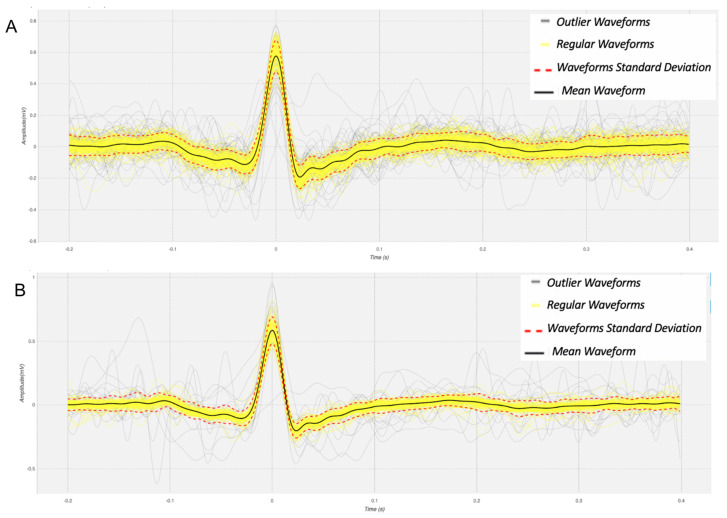
Example heartbeat waveforms for a randomly selected subject, obtained using the electrode materials: (**A**) Keyboard; and (**B**) Keyboard/Mouse.

**Figure 6 sensors-21-07601-f006:**
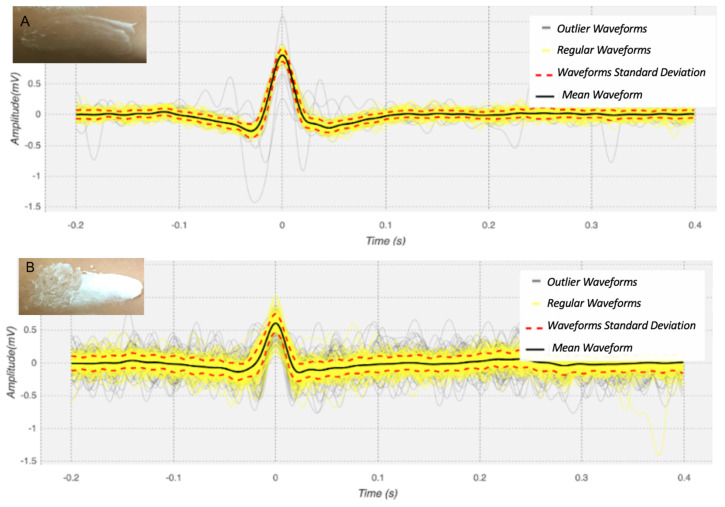
Example heartbeat waveforms for a test subject, obtained with the keyboard electrode when using skin moisturizersith low (**A**) and high density (**B**).

**Table 1 sensors-21-07601-t001:** Impedance between skin and electrode.

**Impedance (kΩ)**	25.0	6.1	6.7	14.8	6.0	53.8	45.2	30.2	18.9	15.6	8.6	6.3
**Frequency (Hz)**	26.51	31.41	44.25	57.74	75.3	80.65	94.34	105.9	136.2	148.3	176.6	193.4

**Table 2 sensors-21-07601-t002:** Comparative analysis of the heart rate values determined for the ECG time series obtained with the reference sensor (ECG REF) and with the experimental sensor (ECG EXP) in Keyboard or Keyboard+Mouse.

Material	QRS (%)	HR (BPM)	ΔHR (BPM)	SDE (%)	*p*-Value
**ECG REF**		78.49 ± 2.55			
**Keyboard**	99.99 ± 1.11	78.00 ± 5.63	0.49 ± 5.05	3 ± 0.01	0.040 ± 0.010
**Keyboard/Mouse**	95.83 ± 3.45	70.00 ± 5.55	8.49 ± 5.14	15 ± 2.04	0.051 ± 0.011

QRS—Percentage of QRS complexes detected with the experimental sensor in relation to the reference QRS; HR—Heart rate (in BPM); ΔHR—Difference between HR detected with the ECG REF and the ECG EXP; EDS—Signal Detection Error; *p*-value—*p*-value of the *t*-test (analysis of the data derived from the signals obtained with the ECG EXP electrodes in relation to the ECG REF electrode).

**Table 3 sensors-21-07601-t003:** Comparative analysis of DFA and Poincaré features for data collected with each electrode material.

	Poincaré				DFA	
Material	SD1 (ms)	SD2 (ms)	S (ms${}^{2}$)	SD1/ SD2	α1	α2
**ECG REF**	71.6 ± 19.1	114.7 ± 30.8	25,830.4 ± 97.3	0.6 ± 29.3	0.9 ± 0.7	1.3 ± 2.7
**Keyboard**	334.8 ± 123.8	438.3 ± 234.8	461,055.4 ± 23.9	0.8 ± 0.8	0.6 ± 0.9	0.8 ± 0.6
**Keyboard/Mouse**	601.8 ± 320.90	582.8 ± 178.9	1,101,947.5 ± 345.0	1.0 ± 1.6	0.9 ± 1.99	1.0 ± 1.4

**Table 4 sensors-21-07601-t004:** Pearson correlation coefficient (PCC) and Normalized Root-mean-square error (NRMSE) between the heartbeat waveforms of ECG REF and ECG EXP.

Material	Keyboard		Keyboard/Mouse	
**Subject**	**PCC**	**NRMSE**	**PCC**	**NRMSE**
1	0.98 ± 0.07	19.86 ± 11.07	0.78 ± 0.30	57.65 ± 25.02
3	0.81 ± 0.14	30.34 ± 16.88	0.59 ± 0.05	26.16 ± 6.53
4	0.95 ± 0.30	51.65 ± 6.42	0.82 ± 0.14	35.06 ± 24.97
5	0.97 ± 0.25	49.89 ± 3.48	0.36 ± 0.07	25.69 ± 3.98
6	0.99 ± 0.00	21.45 ± 3.01	0.88 ± 0.15	26.16 ± 10.71
7	0.88 ± 0.02	23.45 ± 12.83	0.82 ± 0.29	24.75 ± 24.63
μ ± σ	0.91 ± 0.14	31.07 ± 7.8	0.82 ± 0.23	32.19 ± 11.02

**Table 5 sensors-21-07601-t005:** Comparative analysis of the heart rate values for signals obtained with the electrode when low (A), and high (B) density skin moisturizers are applied to the skin.

Material		Moisturizer A	Moisturizer B
**ECG REF**	**HR (BPM) (μ ± σ)**	95.40 ± 7.663	95.15 ± 8.448
	**QRS(%)**	78.87	69.83
**Keyboard**	**HR (BPM) (μ ± σ)**	95.92 ± 8.475	103.09 ± 5.672
	**QRS(%)**	77.03	75.14
	**ΔHR (BPM)**	0.54 ± 0.06	7.94 ± 0.8
	**PCC**	0.95 ± 0.08	0.63 ± 0.19
	**NRMSE**	3.187 ± 0.99	5.467 ± 0.99

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
