# Peer review of "Invisible ECG for High Throughput Screening in eSports"

_sensors, 2021, doi:10.3390/s21227601_

Round 1

Reviewer 1 Report

  1. Data acquisition section should be improved. What it means O1-LED, A1-ECG REF and so on - there is no specification O1, A1 and so on
  2. In section 4.1, test the nappa electrode impedance, not Ag / AgCl
  3. Comments on the t-test are missing (table 2).
  4. The reference to [18] on line 175 leads to a publication in which there is no method described in the paper.
    Similarly, the reference to [8] on line 239 leads to an unrelated publication.
  5. There are duplicate symbols for different parameters (see formula 2 and 6)
  6. Instead of sigma symbol there is a text sigma on page 7 (line 220)
  7. There is an inconsistency between the description of the Figure 4 and their labels 
  8. The labels in Figure 4 are too small
  9. Figure 3 is incomplete.
  10. What was the QRS duration measured? From the figures it seems very small
  11. The references are in the wrong order, moreover, its description is incomplete

Reviewer 2 Report

The authors devise a sensing approach and proposing a proof-of-concept data collection system using a keyboard armrest and mouse, enabling measurements when interacting with the computer. Overall, the paper structure is adequate, even though it needs some adjustments.

  • There are some typo errors. For example, "The 2 section describes ..." it must be "The Section 2 describes ...". Do not forget to use capital letters when making references to Section, Figure, Table. 
  • In the bibliographic review, the authors need to include some research pursuing the same objective, developing other ways to monitor the heartbeat waves.
  • What factors make it difficult to capture data? What were the main reasons for the outliers?
  • It was necessary to define a baseline to use the captured data and compare them in applications such as classification and segmentation of ECG.

Round 2

Reviewer 1 Report

Comments on the answer

ad 6. Has not been corrected (now is line 221)

ad 7. Has not been corrected (see for example F(n) in labels)

ad 8. Has not been corrected. The font size is like in previous version.

ad 10. The question was about QRS not RR interval.

ad 9 Equation (1) shows that the resistor R is in series in the measuring circuit (current measurement). You can't see it from Figure 3.

ad 3. Generating simulation data based on the data in Table 2, we get completely different results than those presented in the article. That is why I asked for a comment on the test results, not a description of what the t-test is for.

ad 4 Unfortunately, I cannot verify this citation

An additional, incomprehensible statement is 0% signal loss (from the abstract). This does not agree with the data presented in Table 2 (SDE 15%)

Round 3

Reviewer 1 Report

Figure 3 is still incorrect. One channel of the oscilloscope is connected to the electrode array and the other to a resistor that is not connected anywhere.

The data in Table 2 is unreliable, thus I asked for a comment on the t-test. p -value is in the range 0-1. As I understand it, in Table 2 the mean values and standard deviation are given. It is not possible for the standard deviation for p - value to reach the value given in Table 2.

Author Response

This manuscript is a resubmission of an earlier submission. The following is a list of the peer review reports and author responses from that submission.